# Osteopontin and Regulatory T Cells in Effector Phase of Allergic Contact Dermatitis

**DOI:** 10.3390/jcm12041397

**Published:** 2023-02-09

**Authors:** Teresa Reduta, Joanna Bacharewicz-Szczerbicka, Anna Stasiak-Barmuta, Tomasz W. Kaminski, Iwona Flisiak

**Affiliations:** 1Department of Dermatology and Venereology, Medical University of Bialystok, Zurawia 14 St., 15-540 Bialystok, Poland; 2Department of Clinical Immunology, Medical University of Bialystok, Waszyngtona 17 St., 15-274 Bialystok, Poland; 3Pittsburgh Heart, Lung and Blood Vascular Medicine Institute, University of Pittsburgh, Pittsburgh, PA 15260, USA

**Keywords:** allergic contact dermatitis, intracellular osteopontin, T lymphocytes

## Abstract

Studies have shown that osteopontin (OPN) and regulatory T cells play a role in allergic contact dermatitis (ACD), but the mechanisms responsible for their function are poorly understood. The study aimed to determine CD4 T lymphocytes producing intracellular osteopontin (iOPN T cells) and assess the selected T lymphocyte subsets including regulatory T cells in the blood of patients with ACD. Twenty-six patients with a disseminated form of allergic contact dermatitis and 21 healthy controls were enrolled in the study. Blood samples were taken twice: in the acute phase of the disease and during remission. The samples were analyzed by the flow cytometry method. Patients with acute ACD showed significantly higher percentage of iOPN T cells compared with healthy controls which persisted during remission. An increase in the percentage of CD4CD25 and a reduced percentage of regulatory T lymphocytes (CD4CD25highCD127low) were also found in the patients with acute stage of ACD. The percentage of CD4CD25 T lymphocytes showed a positive correlation with the EASI index. The increase in the iOPN T cells can indicate their participation in acute ACD. The decreased percentage of regulatory T lymphocytes in the acute stage of ACD may be related to the transformation of Tregs into CD4CD25 T cells. It may also indicate their increased recruitment to the skin. The positive correlation between the percentage of CD4CD25 lymphocytes and the EASI index may be indirect evidence for the importance of activated lymphocytes—CD4CD25 in addition to CD8 lymphocytes as effector cells in ACD.

## 1. Introduction

Allergic contact dermatitis (ACD) is a chronic inflammatory skin disease affecting 15–20% of the general population in the world [1]. Incidence of allergic contact dermatitis due to nickel sulfate, cobalt chloride, and potassium chromate has remained at a high level for many years in most countries and allergy to the new allergens, e.g., isothiazolinone and fragrances, is still increasing [2].

Allergic contact dermatitis is a hypersensitivity reaction of the delayed type. It occurs in the induction and effector phases. The Langerhans cells (in the epidermis), dendritic cells (in the dermis), and keratinocytes play the main role in the induction phase of ACD. An immune response is initiated by direct contact of haptens (low molecular weight compounds up to 500 Da) or protein (500–1000 Da) with the skin. The effector phase develops as a result of secondary exposure to the hapten. The lymphocytes TCD4 and CD8 play a pivotal role in the effector phase. These subpopulations of cells migrate to the area of penetration of the allergen in the skin [3,4]. Mechanisms responsible for the course of ACD remain incompletely defined. Recently, osteopontin and regulatory T cells (Treg) have been shown to play a role in the pathogenesis of allergic contact dermatitis.

Osteopontin (OPN), a phosphorylated acidic glycoprotein composed of 300 amino acids, was discovered in 1986 in osteoblasts in an extracellular matrix. It belongs to the small integrin-binding ligand N-linked glycoproteins (SIBLING). OPN is also referred to as an early T lymphocyte activation protein (Eta-1 molecule) [5,6]. Osteopontin is characterized by strong chemotactic properties for other inflammatory cells: T lymphocytes, monocytes, and dendritic cells. Osteopontin recruits them to the site of the inflammation, and additionally stimulates monocytes to secrete IL-1. This glycoprotein occurs in two isoforms: secreted OPN (sOPN) and intracellular OPN (iOPN). So far, the functions of both isoforms and the interaction between them have not been thoroughly understood. Zhao et al. have shown that macrophages in mice constitutively expressed sOPN. Furthermore, they found that intracellular osteopontin was produced after cell activation through a toll-like receptor [7]. Further investigations have indicated that the sOPN isoform is responsible for the regulation of the immune reaction and development of effector Th1 and Th17 lymphocytes. As a result, sOPN affects cellular immunity during infections and autoimmune and allergic diseases [8,9,10,11,12].

Osteopontin is secreted in physiological and pathological conditions by many cells of the immune system, including T and B lymphocytes [9]. Osteopontin is called a cytokine because it is involved in signal transmission between cells and participates in inflammatory reactions [13]. Recent studies have shown the role of osteopontin in some skin diseases in which T lymphocytes are involved. The increase of OPN expression in skin lesions and serum from patients with psoriasis has been shown [14,15]. Furthermore, high plasma osteopontin concentrations have been recognized as a risk factor for the development of cardiovascular diseases in patients with psoriasis [16]. Increased plasma OPN concentrations have also been found in patients with sarcoidosis [17] and alopecia areata [18]. 

There is very limited data discussing the role of osteopontin in allergic diseases. The studies of Akelma et al. have shown that the sOPN in serum was increased in children with asthma compared to the healthy group [19]. Similar results were obtained in murine contact hypersensitivity (CHS) reactions induced by transdermal dust mite antigens delivery. The osteopontin secreted by fibroblasts stimulated keratinocytes to produce pro-inflammatory cytokines [20]. 

Only a few studies in recent years have indicated a role of osteopontin in the pathogenesis of both induction and effector phase of allergic contact dermatitis. Depriving of OPN secretion in mice resulted in reduced infiltration of CD4 and CD8 effector lymphocytes in the skin lesions and the symptoms of chronic eczema were less severe [21,22]. Studies have shown that osteopontin affects the differentiation of CD4 and CD8 T lymphocytes [11,22,23,24]. The increased expression of sOPN in the ACD skin lesions [22] as well as in serum of ACD patients have been found [25]. 

Regulatory T cells (Tregs) are the subpopulations of T lymphocytes. The markers of these cells are CD25 (IL-2 receptor alpha chain) and Foxp3 (forkhead box p3 gene product), which play a role in their activity [26,27,28,29]. The CD4CD25 T lymphocytes occur in humans in peripheral blood, thymus, spleen, tonsils, and umbilical cord blood [30]. The CD4CD25Foxp3 phenotype of Tregs constitutes 5–10% of CD4 T lymphocytes in peripheral blood, but transient Foxp3 and CD25 expression can occur also on activated non-Tregs lymphocytes. Therefore, other markers have been also used to identify Tregs cells. The phenotype of Tregs has been also defined as CD4 T cells with very high expression of the CD25 and Foxp3 markers [31]. The CD4CD25highFoxp3 cells constitute 1–2% of CD25 T lymphocytes. Another important marker is CD127, which is a receptor for IL-7, the cytokine that stimulates maturation and differentiation of T lymphocytes in the thymus. The CD127 expression on the Tregs cells is inversely correlated with Foxp3 expression. Cytoplasmic expression of Foxp3 has also been demonstrated in cells showing no suppressor activity, such as activated CD4CD25 cells [32]. The finding of CD25 high expression and CD127 low expression allows the detection of approximately 98% of regulatory T cells. [33,34]. 

Regulatory T cells have been found in the human skin, where they constitute 20% of CD4 T lymphocytes. The majority of Tregs in the skin such as Tregs in peripheral blood are characterized by low CD127 expression. The circulation of the Tregs from peripheral blood to the skin is poorly understood [35,36,37,38,39].

The regulatory T cells participate in the maintenance of the tolerance to self-antigens. They also suppress the immune response by direct inhibition of the activation, proliferation, and function of CD4 and CD8 effector cells [40]. Additionally, Tregs regulate the inflammatory response [41,42]. 

Studies have shown that regulatory T cells play a role in allergic diseases, including allergic contact dermatitis. In recent investigations, the role of regulatory T lymphocytes in ACD has been found, however, the function and mechanisms by which the Tregs regulate the inflammatory process in the skin are poorly understood. 

The study aimed to determine CD4 T lymphocytes producing intracellular osteopontin and assess the T lymphocyte subsets CD4, CD4CD25, CD4CD25high, and CD4CD25highCD127low in the blood of patients with allergic contact dermatitis.

## 2. Material and Methods 

The study included twenty-six adult patients (11 men and 15 women) with a disseminated form of allergic contact dermatitis and 21 healthy controls (9 men and 12 women). The diagnosis of allergic contact dermatitis was established based on a detailed medical history of patients, the physical examination, and positive result of patch tests. We analyzed the duration of current eczematous lesions, the coexistence of atopy, and other inflammatory diseases in which the elevated serum OPN level has been reported (Table 1). 

The patients with a disseminated form of allergic contact dermatitis (at least two lesions of contact dermatitis in different localizations) in the acute phase of the disease were qualified for the study. The extent and severity of skin lesions were assessed by the EASI index (eczema area and severity index). The EASI index has been described by Hanifin [43] for the evaluation of skin lesions in atopic dermatitis and adopted for the assessment of the disseminated form of allergic contact dermatitis [25]. The EASI scoring system includes the clinical symptoms (erythema, infiltration, papules, erosions, and lichenification) in a particular localization: head, trunk, upper and lower limbs. Additionally, the EASI index determinates the percentage of skin area affected by ACD lesions in each region. Patients with allergic contact dermatitis were classified into two groups according to the severity of the skin lesions: mild to moderate (EASI < 15) and severe (EASI > 15) course of the disease. The patients did not have any clinical symptoms of bacterial skin nor general infection and the level of CRP was normal. The patients were not treated with oral glucocorticoids for a minimum of 3 weeks before enrolling in the study and they did not use any other immunosuppressive or antihistamine drugs. 

Peripheral blood T lymphocytes producing intracellular osteopontin and CD4, CD4CD25, CD4CD25high, and CD4CD25highCD127low T cells were examined in the group of 26 patients with disseminated allergic contact dermatitis (15 women and 11 men). The age of the patients was from 19 to 73 (mean 47.5 ± 14.9) years. The control group consisted of 21 healthy people: 12 women and 9 men, aged 22 to 74 (mean 45.8 ± 15.1). The duration of current skin lesions ranged from 1 to 12 (mean 3.7 ± 3.2) weeks. The EASI index ranged from 2.0 to 29.6 (mean 11.6 ± 7.3). The atopic diseases (asthma, atopic dermatitis, conjunctivitis, and rhinitis) were identified in four patients (3 women and 1 man); other diseases (chronic obstructive pulmonary disease, hypertension, and hypercholesterolemia) were found in 8 patients (5 women and 3 men). Positive results of patch tests with metals were found in 16 patients and with other allergens in 10 patients. 

In patients with allergic contact dermatitis, the blood samples were taken twice: in the acute phase of the disease before treatment and during the remission. After taking the first blood sample, all patients were treated with topical glucocorticoids. Additionally, four patients received oral glucocorticoids. 

The research was approved by the Local Bioethical Committee of Medical University in Bialystok and was in accordance with the Helsinki Declaration.

### 2.1. Evaluation of Peripheral Blood Morphology

The venous blood sample of 2 mL was collected from the patient into a tube with EDTA for evaluation of the leukogram components. The leukogram analysis was performed on a hematological analyzer (Sysmex, Japan). The leukocyte count was given in 10^9^/L, and the lymphocytes were presented in absolute (BL in 10^9^/L) and percentage (%) values. At the beginning of the study in ACD patients and the control group, the leukocyte count, and the absolute and percentage lymphocyte values were assessed.

### 2.2. The Evaluation of T Lymphocytes with Expression of Intracellular Osteopontin in Peripheral Blood by the Method of Flow Cytometry

Each blood sample was divided into 50 μL aliquots, then 10 μL of CD4-PC5 conjugated antibody (Beckman Coulter, Pasadena, CA, USA) was added to each aliquot. After 15 min of incubation at room temperature, 100 μL of cell membrane fixing reagent A (IntraStain-Dako, Glostrup, Denmark) was added to each sample. The blood samples were incubated again for 15 min and then the cell suspension was washed 3 times in phosphate-buffered saline (PBS). Additionally, to each sample, 100 μL B membrane permeabilization reagent (IntraStain-Dako) and 20 μL of anti-Osteopontin (R&D) antibodies were added. After another 15 min of incubation in the darkroom at room temperature, the cell suspension was washed again 3 times in phosphate-buffered saline. To each sample, 250 μL PBS and 50 μL 1% paraformaldehyde were added. After thorough mixing, the samples were analyzed by the flow cytometry method (Cytomics FC 500, Beckman Coulter). 

### 2.3. The Evaluation of CD4, CD4CD25, CD4CD25high, and CD4CD25highCD127low T Lymphocytes in Peripheral Blood by the Method of Flow Cytometry

The CD4 T lymphocytes were assessed in peripheral blood by the method of flow cytometry. Ten μL of monoclonal antibodies CD4-PC5, CD25-FITC, and CD127-PE (Beckman Coulter) were added to each sample with 100 μL of whole blood. After 30 min of incubation at room temperature, each sample was subjected to a rapid, automatic lysis process (ImmunoPrep Work Station, Beckman Coulter). After thorough mixing, the samples were analyzed by the method of flow cytometry (Cytomics FC500, Beckman Coulter) and every 10^5^ cells were counted. The control was determined according to the FMO (fluorescence-minus-one) method. 

### 2.4. Statistical Analysis 

Normality of distribution of data was assessed using Shapiro-Wilk test. The results meeting the Gaussian distribution assumption were shown as mean ± SD (standard deviation), while non-Gaussian results are presented as median with the ranges (minimum–maximum values). The Student *t* test (parametric) or Mann-Whitney (non-parametric) tests were used to compare differences between two data sets. To assess the differences between more than 2 subgroups, analysis of variance (ANOVA) test was used. The correlations were analysed using Spearman’s rank correlation analysis or Pearson correlation, based on the data set distribution. A two-tailed *p* value < 0.05 was considered statistically significant. Computations were performed using GraphPad 8 Prism (GraphPad Software; La Jolla, CA, USA).

The study was conducted in accordance with the Declaration of Helsinki and approved by the Ethics Committee of Medical University of Bialystok (No: R-I-002/140/2012), Poland.

## 3. Results

There was no difference between absolute and percentage value of T lymphocytes; only interest rates were used for further calculations. The examined populations of CD4 T lymphocytes in the blood of ACD patients and healthy controls are presented in Figure 1.

The percentage of T lymphocytes with iOPN expression was significantly higher in patients with the acute phase of ACD than in the patients in remission (*p* < 0.01) and healthy group (*p* < 0.0001). During the remission of ACD, the percentage of CD4 lymphocytes with intracellular osteopontin remained significantly higher than in the healthy group (*p* < 0.0001). The percentage of CD25 and CD25h T lymphocytes was significantly higher in patients with the acute phase of disease comparing to patients with remission (*p* < 0.0001 and *p* < 0.05, respectively) and controls (*p* < 0.0001). The percentage of CD25hCD127low T lymphocytes was significantly lower in the acute phase of ACD comparing to controls (*p* < 0.05). The correlation between iOPN CD4 cell percentage and selected variables in patients with ACD is presented in Table 2.

The percentage of T lymphocytes with iOPN expression in patients with acute phase and remission of ACD did not correlate with patients’ age, duration, and severity of skin lesions (EASI) and with the percentage of all examined CD4 T cell populations. However, during remission, the correlation coefficient between iOPN CD4 and CD25h was 0.338 (*p* = 0.092).

The percentage of iOPN CD4 T lymphocytes and other T cell subpopulations according to the duration of skin lesions (<1 week and >1 week) are presented in Figure 2. 

In all ACD patients with lesions lasting < 1 week and with lesions lasting > 1 week both in the acute stage and in remission, the percentage of iOPN CD4 T cells was significantly higher than in the control group (*p* < 0.0001). Additionally, patients with a short duration of skin lesions showed a statistically higher percentage of iOPN CD4 T cells in the acute phase of ACD than in remission (*p* < 0.05). In patients with the acute disease regardless of the duration of skin lesions, the percentages of CD25 and CD25h populations were statistically higher than in the controls (*p* < 0.0001 and *p* < 0.05, respectively). Furthermore, in patients with skin lesions lasting more than 1 week, the percentage of CD25 during remission remained statistically higher than in the healthy group. In patients with a short duration of ACD lesions, the level of CD25 and CD25h was in the acute stage significantly higher than in remission (*p* < 0.05 and *p* < 0.05, respectively). There were no statistically significant differences in the proportion of CD4CD25hCD127low cells between the two groups depending on the duration of skin lesions. 

The percentages of iOPN CD4 T cells and other T lymphocyte populations according to the severity of ACD are presented in Figure 3. 

All patients regardless of the extent of skin lesions presented a significantly higher level of T lymphocytes with iOPN expression than the healthy group (*p* < 0.0001). This difference was observed in two EASI groups of patients in both acute stage and remission of ACD (*p* < 0.0001 and *p* < 0.01, respectively). In patients with EASI < 15 in the acute phase of the disease, the level of iOPN CD4 T cells was significantly higher than in remission (*p* < 0.05). In both EASI groups percentage of iOPN CD4 T cells did not correlate with the age of the patients, duration of skin lesions, EASI score, and with proportion of each examined T cell population (Table 3A,B).

In patients in remission with more severe skin lesions (EASI > 15), the values of iOPN CD4 T lymphocytes and CD25h showed correlation coefficiency near the significance threshold (*p* = 0.061) (Table 3B). The percentage of CD4 T lymphocytes in patients with mild to moderate skin lesions did not differ between all groups. In patients with EASI > 15, the percentage of CD4 T cells was significantly higher during the acute stage of disease compared to remission (*p* < 0.05) and to the healthy group (*p* < 0.05). The proportion of CD25 T cells was significantly higher in the acute stage of ACD in both groups (EASI < 15 and EASI > 15) than in controls (*p* < 0.01). In patients with mild to moderate skin lesions (EASI < 15) examined in the acute phase of ACD, significantly higher percentage of CD25 T lymphocytes was observed comparing to remission (*p* < 0.01). A higher proportion of CD25h T cells was also observed in patients with acute ACD of both EASI groups; the difference was not statistically significant. During acute disease, patients showed a lower proportion of CD4CD25hCD127low than the control group. In a group with EASI < 15 this difference was significant (*p* < 0.01). 

In patients lacking comorbidities, the percentage of iOPN CD4 cells was significantly higher in the acute ACD stage than in remission (*p* < 0.05). The proportion of CD4 population did not differ between patients lacking comorbidities and with comorbidities. In patients without comorbidities, the percentage of CD25 T cells was statistically higher in acute stage than in remission (*p* < 0.0001). A higher proportion of CD25h T cells was also observed in patients lacking comorbidities in the acute stage of ACD, but the difference was not statistically significant (*p* = 0.074). In patients without comorbidities, the CD25hCD127low population was statistically higher in remission than in the acute stage of ACD (*p* < 0.05) (Figure 4). 

Comparison of selected variables in ACD patients depending on the presence or lack of comorbidities (Table 4A) showed that in patients with coexistence of other diseases the proportion of CD25 cells was in acute stage significantly higher than during remission (*p* < 0.05). Additionally, in patients without comorbidities, the percentage of CD25 was significantly higher in the acute stage than in remission (*p* < 0.0001). The subpopulation of CD25hCD127low was higher in patients lacking comorbidities during the acute stage of disease than in remission; the difference was near the significance threshold (*p* = 0.07) (Table 4A). Correlation of the percentage of iOPN CD4 lymphocytes with selected variables in ACD patients depending on the presence of comorbidities is presented in Table 4B. In patients with other diseases, there was a positive correlation between the percentage of iOPN CD4 T lymphocytes and proportion of CD4 (*p* = 0.019). Proportion of CD25 and CD25h correlated with iOPN CD4 percentage in patients lacking comorbidities examined during remission (*p* = 0.042 and *p* = 0.034, respectively).

In patients with allergy to metals, the level of iOPN CD4 T cells was statistically higher in the acute stage than in remission (*p* < 0.05). The proportion of the CD4 population did not differ in patients with allergies to metals and other allergens. In patients with allergy to metals, the CD25 T cells were statistically higher in the acute stage of ACD than in remission. In patients without allergy to metals, the percentage of CD25hCD127low was statistically higher in remission of ACD than in the acute stage of the disease (Figure 5). 

The comparison of selected variables for patients with ACD depending on allergen is presented in Table 5A. In patients allergic to metals, during the acute stage proportion of CD25 was significantly higher than in remission (*p* < 0.01). In patients allergic to other allergens, during the acute stage the proportion of CD25h was significantly higher than in remission (*p* < 0.05) (Table 5A). Correlations of the percentage of iOPN CD4 T lymphocytes with selected variables in the ACD patients depending on allergen are presented in Table 5B. There was a positive correlation between iOPN CD4 percentage and CD25h in patients allergic to metals examined during remission (*p* < 0.01) (Table 5B).

## 4. Discussion

The biological role of intracellular osteopontin is not fully understood, in contrast to the function of sOPN. Intracellular osteopontin is located around the nucleus and near the cell membrane within the cell. Studies have shown the role of iOPN in cell motility, rearrangement of the cytoskeleton, and the mitosis process. Additionally, iOPN affects the function of the immune cells in viral or fungal infections and neoplasms [44,45,46,47,48,49,50]. 

A few studies on the role of osteopontin in the murine model of allergic contact dermatitis were performed. Results of one of the studies showed the increased osteopontin expression in the dermis-subpapillary plexus and the regional lymph nodes during the induction phase after contact with allergens [23]. Results of recently published research revealed that osteopontin induced migration of the Langerhans cells and dendritic cells from epidermis and skin to the lymph nodes, where it affected the polarization of the response toward Th1 cells [23,24,51]. Only a few studies have shown the role of osteopontin in the effector phase of ACD. In the murine CHS model, the increased expression of sOPN on effector T cells infiltrating inflammatory eczematous skin lesions was identified. [22]. Mild ear swelling after contact with an allergen was found in mice lacking osteopontin secretion. It was correlated with impaired ability to migrate Langerhans cells to the lymph nodes [23]. Researches revealed increased sOPN level in serum in patients with chronic phase of ACD [22] and the disseminated, acute form of allergic contact dermatitis [25,52]. The increase of sOPN level in serum was seen especially during the first two weeks of the disease and it was maintained by the duration activity of skin lesions. The concentrations of sOPN in serum were not different in patients with the remission of ACD and in the healthy group. It can indicate that sOPN is secreted only in a period of active skin lesions. It has not been established whether the source of sOPN in the serum is lymphocytes or other cells present in the blood or in the skin [25].

The knowledge about the role of the intracellular osteopontin isoform in ACD is limited. Therefore, this study assessed the expression of iOPN in peripheral blood CD4 T lymphocytes in patients with allergic contact dermatitis. 

The study showed that the percentage of T lymphocytes with iOPN expression was significantly higher in patients with the acute phase of ACD than in the patients in remission (*p* < 0.0001) and the healthy group (*p* < 0.0001). During the remission of ACD, the percentage of CD4 lymphocytes with intracellular osteopontin remained significantly higher than in healthy persons (*p* < 0.01). The presented study showed convergence with the results of other studies showing an increase of sOPN in the serum of patients with acute stage of allergic contact dermatitis [25]. A simultaneous increase in the percentage of iOPN CD4 T lymphocytes and serum OPN concentration in patients with acute stage of ACD may indicate the origin of sOPN from blood lymphocytes. A significant increase in the iOPN CD4 T cells’ percentage during remission could indicate the persistence of the increased number of these T lymphocytes in the blood, despite the disappearance of skin lesions without the release of osteopontin from T cells.

The pathogenesis of allergic contact dermatitis involves selected T cell subpopulations: CD4, CD4CD25, CD4CD25high, and regulatory T cells defined as CD4CD25highCD127low. Therefore, these were assessed in this study. 

In allergic contact dermatitis, the role of effector T cells is mainly attributed to CD8 T lymphocytes. The CD4 population is characterized by the ability to both stimulate and inhibit inflammation. In murine CHS reactions, CD4 T cells play a main role in inhibiting inflammatory response after contact with haptens [53]. However, CD4 T lymphocytes can play an effector role during the absence or deficiency of the CD8 population. In dinitrochlorobenzene-sensitized mice, it has been shown that CD8 lymphocytes migrate to the skin first. Then effector CD4 T cells appear at the site of inflammation and they synthesize IFN-γ [54]. Regulatory lymphocytes, as well as effector lymphocytes, belong to the T lymphocytes and constitute 5–10% of peripheral blood CD4 T cells. These cells are involved in inhibiting the inflammatory response. The role of regulatory lymphocytes has been proven in some inflammatory skin diseases, including psoriasis, alopecia areata, and atopic dermatitis. However, previous studies have not provided conclusive results. Patients with psoriasis showed a reduced number and impaired suppressive function of Treg in the blood and skin which correlated with disease exacerbation [55,56,57]. In patients with alopecia areata, impaired activity of Treg cells has been shown, with no effect on their number. It was found that circulating Treg lymphocytes were unable to inhibit the activated cytotoxic lymphocytes involved in the development of this disease [58,59]. However, Kubo et al. showed an increased number of circulating TCD4CD25 lymphocytes in patients with alopecia areata compared to the control group [60]. In patients with atopic dermatitis, an increased number of circulating CD4CD25Foxp3 T cells was found in the blood compared to the analogous value assessed in the healthy group [61,62,63]. Different results were presented by Szeregi et al. which did not show differences in the number of CD4CD25Foxp3 lymphocytes in patients with atopic dermatitis and the healthy group [64]. 

The evaluation of the importance of Treg in ACD was initiated by a study of Cavani showing that nickel poorly influenced the proliferation of TCD4 lymphocytes in the blood of healthy persons. However, depriving of the CD4 and CD4CD25 populations lead to its intensification [65]. The study confirmed that CD25 T cells inhibited effector cells in allergic contact dermatitis. In the normal skin and nickel-coated skin in patients with allergy to nickel and the healthy group, the study assessed also the infiltration of T lymphocytes. The CD25 T cells’ infiltration was not found in the skin of allergic patients not exposed to nickel. On the other hand, in non-sensitized individuals, no signs of inflammation were present at the nickel application site, but an increased infiltration of CD25 T lymphocytes was found in the skin specimen. This shows the migration of CD25 T lymphocytes to the site of contact with the allergen and active inhibition of the immune reaction by these cells in healthy individuals [65,66]. The present study showed that the percentage of CD25 T lymphocytes was significantly higher in patients with acute stage of ACD comparing to patients with remission (*p* < 0.0001) and controls (*p* < 0.0001). The CD25 molecule is one of the markers of regulatory T lymphocytes, but it is also an expression of the activation of effector lymphocytes. The results of this study indicate the activation of CD4 lymphocytes in the acute stage of the disease. It confirms their participation in the pathogenesis of the acute stage of allergic contact dermatitis as effector cells. The results of the study also showed a statistically higher value of CD25h lymphocytes in the acute stage of the disease compared to patients in remission and the control group (*p* < 0.05). The obtained results of the study could indicate that a part of the CD25 T cell population constitutes both effector and regulatory lymphocytes. 

In this study, Treg cells were characterized by the presence of CD4 high expression of CD25 receptors and low or no expression of the CD127 receptor. In patients with the acute phase of allergic contact dermatitis, the percentage of CD4CD25highCD127low T cells was significantly lower than in the healthy group (*p* < 0.05). The decreased number of regulatory T lymphocytes in the blood of patients in the acute stage of the disease may be one of the reasons for the clinical manifestation of an allergic reaction. The likely cause of the observed phenomenon may be the increased recruitment of Tregs to the the inflammatory reaction of the skin at the site of allergen challenge. The process of passing of the Treg cells from circulation to the skin is poorly understood. Recently, it has been shown that in mice, T cells with TCR receptors massively flow from the thymus to the skin between 6 and 13 days postnatally. Among these lymphocytes as much as 80% are activated Treg cells [36]. Previous studies have shown the presence of the cutaneous lymphocyte antigen (CLA) on 68–90% of CD4CD25highFoxp3 lymphocytes in peripheral blood of healthy adults. This antigen is responsible for Treg migration to the skin [67]. Studies assessing the migration of CD4 T lymphocytes in the inflammatory skin reaction in mice have shown that these cells migrate along collagen fibers with the participation of integrins [68]. Regulatory lymphocytes in the skin show less mobility in comparison with effector lymphocytes [37]. The antigen-specific memory Treg lymphocytes have been also found in the skin. They persisted in the skin long after antigen removal and resolution of inflammation. Upon antigen re-exposure, these cells were responsible for suppressing the inflammation. This response was faster than the primary one observed after the first antigen administration [69].

In the further part of the presented research, the analysis of examined T lymphocyte subpopulations depending on the duration of skin lesions, EASI index, the coexistence of other inflammatory diseases, and the type of allergen was carried out.

In the presented study, it was found that the percentage of iOPN T lymphocytes showed statistically higher values both in the acute and remission stages regardless of the duration of skin lesions as compared to the healthy group. The highest percentage of iOPNCD4 T lymphocytes was observed in patients with skin lesions lasting less than a week. The results could indicate that in patients with ACD, osteopontin could show its particular contribution in the initial stage of ACD and may be used as an indicator of disease activity. The increase in the percentage of iOPN T lymphocytes in the early stage of the disease corresponds to the results of studies by Weis et al. They showed an increased concentration of sOPN in lymph nodes shortly after the application of the allergen to the skin of mice [23]. Higher levels of secreted osteopontin have also been found in the serum of patients with skin lesions lasting less than two weeks [25].

The study showed that the percentage of CD4CD25 T lymphocytes was significantly higher in the acute stage of the disease, regardless of its duration, compared to the control group. This confirms the participation of this population as effector cells in ACD. In patients with skin lesions lasting more than one week, the number of CD25 lymphocytes remained elevated during remission, which indicates that despite the disappearance of clinical symptoms, an increased pool of effector lymphocytes remains in the blood without the development of skin lesions. In patients with skin lesions lasting less than a week, an increase in the CD25h T lymphocytes percentage was also observed compared to the control group. However, this increase was not as pronounced as in the case of CD25 cells. It could indicate that some of them constitute the population of CD4CD25highCD127low lymphocytes. The previous investigation revealed in ACD patients with skin lesions lasting up to one week the lowest percentage of CD25 and CD25h T lymphocytes. In patients with skin lesions lasting 2 weeks, the percentage of CD25 cells increased significantly; in patients examined after 3 weeks the increase of CD25h cells was also observed [70]. In all groups of patients with ACD presented in our study in the acute stage of the disease, there was a significant reduction in the percentage of regulatory T lymphocytes in the blood. The lowest values were observed in patients with skin changes of short duration. These differences were however not significant, most likely due to the small study group. 

The study showed that regardless of the severity of skin lesions (EASI < 15 and EASI > 15), the percentage of CD4iOPN T lymphocytes was statistically higher both in the acute stage and in the remission of ACD compared to the healthy group. In the presented study, no correlation was found between the iOPN T lymphocyte subpopulation and the severity of skin lesions. In the studies by Reduta et al., serum sOPN concentration was significantly higher in patients with severe skin lesions, which could indicate different sources of the two forms of osteopontin in the blood. It is difficult to determine the origin of sOPN in the blood in ACD, and the source of the increased OPN concentration in serum found in patients with ACD may be circulating lymphocytes, monocytes, or other cells. It is not known yet whether iOPN can be converted into sOPN, and it cannot be ruled out that the source of sOPN is the skin involved with inflammation. On the other hand, the results presented above suggest that sOPN may be derived at least in part from circulating effector T lymphocytes. Further investigation would be required to confirm this fact. 

In patients with severe skin lesions studied in the acute stage of the disease, the CD4 T lymphocytes’ percentage was significantly higher (*p* < 0.05) than in the healthy group and in remission. This is most likely due to the increase in the number of circulating effector T lymphocytes. These differences were not visible in the whole group of patients. Regardless of the EASI, the percentage of the CD25 T lymphocytes was significantly higher in acute ACD, both concerning the healthy group and patients in remission. The differences in the degree of significance most likely resulted from the size of the studied groups. This would require confirmation in larger groups of patients. Despite a significant increase in CD25h T lymphocytes in the whole group of patients with acute ACD as compared to those in remission and the control group, this difference was not visible in individual EASI-dependent groups. This is probably related to the small number of people in these groups. The lower percentage of CD4CD25highCD127low cells found in the acute stage of the disease was especially noticeable in the group of patients with less severe lesions, which is probably due to the small group of patients with severe skin lesions. In groups of patients with severe lesions, it was found that the correlation coefficient between iOPNCD4 and CD25h was at the threshold for statistical significance. Probably in a larger group of patients the correlation would be more visible, which could suggest that CD4iOPN T lymphocytes belong to the CD25h T cells (Table 3A,B). 

The relationship between the expression of iOPN in T lymphocytes and the coexistence of systemic diseases with increased secretion of osteopontin was also assessed in the studied patients. Previous studies have shown an increased concentration of osteopontin in the serum in the course of atherosclerosis, chronic obstructive pulmonary disease, and diabetes mellitus [71,72,73]. In both study groups, there were higher values of the percentage of iOPN T lymphocytes in the acute stage of ACD than in the remission. In the group of patients without comorbidities, this difference was significant, which excluded the influence of comorbidities on the obtained result. However, patients with comorbidities constituted only 7.7% of the control group, making the result inconclusive. The result should be checked on a larger number of patients. Moreover, patients with a history of the above diseases were examined during the remission and the results of laboratory tests were normal. It has also been shown that the statins taken by the studied patients are responsible for lower concentration of OPN, which may be related to the observed lack of differences in the obtained results [74]. Similar results were observed in patients without comorbidities with regard to CD25 and CD25h T lymphocytes. In the case of CD25 lymphocytes, this difference was statistically significant (*p* < 0.0001). This difference can indicate that the lack of other diseases excludes their influence on the result and supports their significance in the development of eczema skin lesions. 

At the end of our research, we analyzed the examined subpopulations of T lymphocytes depending on the type of allergen. In the acute stage of the disease, the percentage of CD4iOPN T lymphocytes was higher independently of the type of allergen; in the case of metals, the difference was statistically significant (*p* < 0.05). It is difficult to explain the reason for this difference; maybe it is due to a smaller group of patients with non-metal allergy, or it may be due to a stronger stimulation of T cells to produce iOPN by metals, which have been shown to have the ability to directly stimulate T cells (Figure 5). In the studies of Seier et al., it was found that in patients with allergy to nickel, CD4 and CD8 T lymphocytes secreted increased amounts of osteopontin [22]. Regardless of the type of allergen, the percentage of CD25 was higher in the acute stage of the disease. In the group with an allergy to metals, the difference was significant (*p* < 0.01), and in the remaining patients it was near the *threshold* of statistical *significance.* This probably resulted from the smaller group of these patients. In the case of allergy to non-metal haptens, such a relationship was observed in the population of CD25h lymphocytes, which could indicate the participation of these cells in sensitization to this group of allergens. In both groups of patients, depending on the allergen, the percentage of regulatory lymphocytes was reduced in the acute stage of the disease, which could indicate their role in inhibiting ACD inflammation, regardless of the type of allergen.

## 5. Conclusions

The increase in the T cell population with intracellular expression of osteopontin can indicate its participation in acute ACD. Taking into account previous studies showing an increase sOPN level in the serum of patients with acute ACD could indicate the origin of sOPN from blood lymphocytes and the possibility of transformation of iOPN into sOPN.These results can also confirm the pro-inflammatory effect of OPN in acute ACD. An increased percentage of T lymphocytes with iOPN expression during remission of skin lesions can indicate the prolonged presence of these cells in the blood, however, there is no OPN secretion from T lymphocytes.The decreased percentage of regulatory T lymphocytes in the blood of patients with acute ACD in comparision with results in the healthy group may be related to the transformation of Tregs into CD4CD25 T cells, of which increased percentage was found in the acute stage of contact dermatitis. It may also indicate their increased recruitment to the skin.The positive correlation between the percentage of CD4CD25 T lymphocytes and the EASI index may be indirect evidence for the importance of activated lymphocytes with the CD4CD25 phenotype (in addition to CD8 lymphocytes) as effector cells in allergic contact dermatitis.

## Figures and Tables

**Figure 1 jcm-12-01397-f001:**
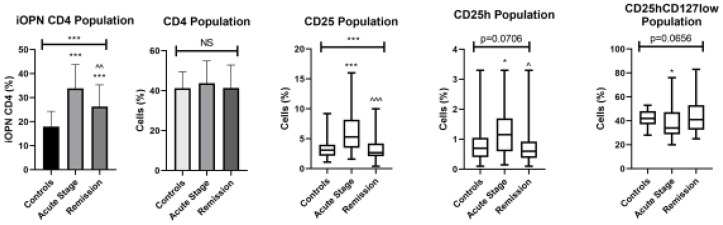
The examined populations of CD4 T lymphocytes in the patients with ACD and healthy controls. iOPN CD4, CD4 T lymphocytes with expression of intracellular osteopontin; CD4, CD4 positive T lymphocytes; CD25, subset of CD4 positive T lymphocytes; CD25h, T lymphocytes with high CD25 expression; CD25hCD127low, T lymphocytes with high CD25 expression and CD127 low expression—regulatory T cells; NS, non-significant; * means statistical significance with *p* values < 0.05 between patients with acute stage of ACD and controls; *** means statistical significance with *p* < 0.0001 between patients with acute stage of ACD and controls; ^—statistical significance with *p* values < 0.05 between patients with acute stage of ACD and remission; ^^/^^^—statistical significance with *p* values < 0.01 and 0.0001, respectively, between patients with acute stage of ACD and remission.

**Figure 2 jcm-12-01397-f002:**
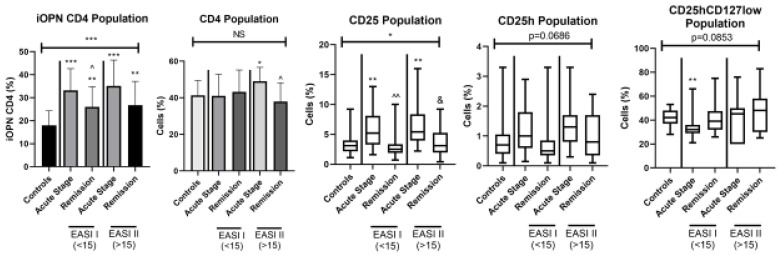
The percentage of selected T lymphocytes according to severity of ACD. iOPN CD4, CD4 T lymphocytes with expression of intracellular osteopontin; CD4, CD4 positive T lymphocytes; CD25, subset of CD4 positive T lymphocytes; CD25h, CD25 high T lymphocytes; CD25hCD127low, CD25highCD127low T cells—regulatory T cells; NS, non-significant; EASI I (<15), Eczema Area and Severity Index in patients with mild to moderate course of the disease; EASI II (> 15), Eczema Area and Severity Index in patients with severe course of the disease; * means statistical significance with *p* values < 0.05 between patients with acute stage of ACD and controls; **/*** means statistical significance with *p* < 0.01 and 0.0001, respectively, between patients with acute stage of ACD and controls; ^/^^ means statistical significance with *p* values < 0.05 and < 0.01, respectively, between patients with acute stage of ACD and remission; & means statistical significance with *p* values < 0.05 between patients with remission and controls.

**Figure 3 jcm-12-01397-f003:**
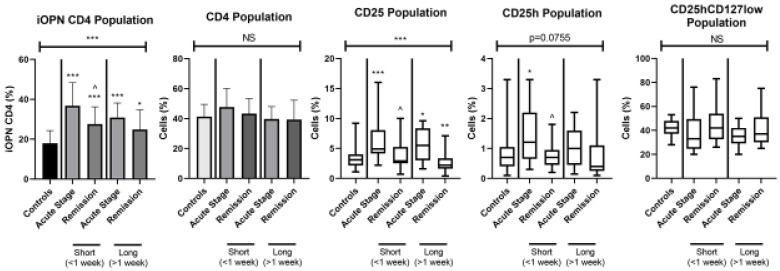
The examinated subpopulations of T lymphocytes in the ACD patients according to duration of skin lesions. iOPN CD4, CD4 T lymphocytes with expression of intracellular osteopontin; CD4, CD4 positive T lymphocytes; CD25, subset of CD4 positive T lymphocytes; CD25h, CD25 high T lymphocytes; CD25hCD127low, CD25highCD127low T cells—regulatory T cells; NS, non-significant; short, the duration of skin lesions <1 week; long, the duration of skin lesions >1 week; * means statistical significance with *p* values < 0.05 between patients with acute stage of ACD and controls; **/*** means statistical significance with *p* < 0.01 and 0.0001, respectively, between patients with acute stage of ACD and controls; ^—statistical significance with *p* values < 0.05 between patients with acute stage of ACD and remission.

**Figure 4 jcm-12-01397-f004:**
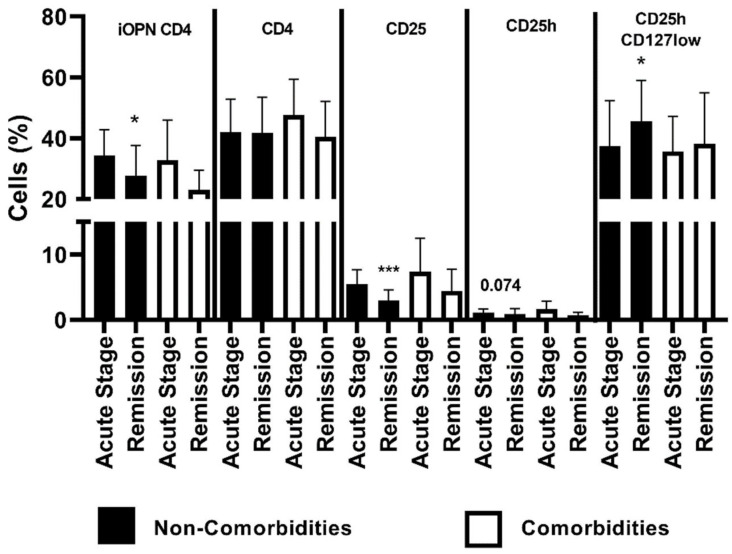
The percentage of examined populations of CD4+ T lymphocytes depending on the presence of comorbidities. iOPN CD4, CD4 T lymphocytes with expression of intracellular osteopontin; CD4, CD4 positive T lymphocytes; CD25, subset of CD4 positive T lymphocytes; CD25h, CD25 high T lymphocytes; CD25hCD127low, CD25highCD127low T cells—regulatory T cells; NS, non-significant; * means statistical significance with *p* values < 0.05 between patients with acute stage of ACD and with remission; *** means statistical significance with *p* < 0.0001 between patients with acute stage of ACD and with remission.

**Figure 5 jcm-12-01397-f005:**
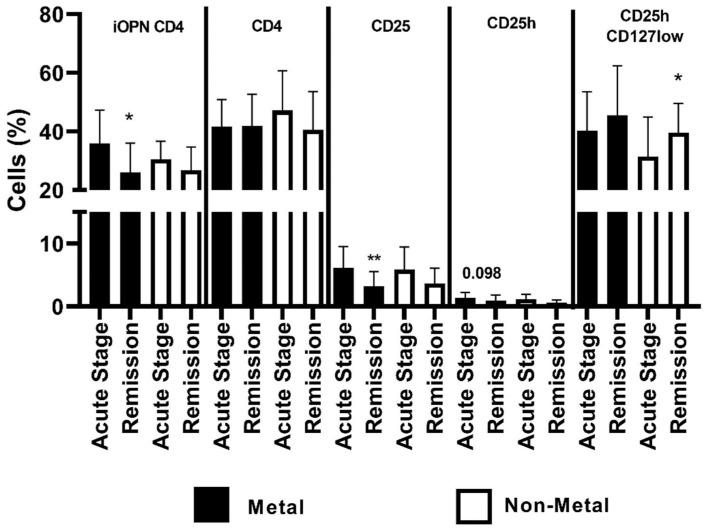
The percentage of examined populations of CD4 T lymphocytes depending on the type of allergen. iOPN CD4, CD4 T lymphocytes with expression of intracellular osteopontin; CD4, CD4 positive T lymphocytes; CD25, subset of CD4 positive T lymphocytes; CD25h, CD25 high T lymphocytes; CD25hCD127low, CD25highCD127low T cells—regulatory T cells; *, means statistical significance higher the percentage of iOPN CD4 with *p* values < 0.05 between patients in acute stage of ACD than in remission; **, means statistical significance higher the percentage of CD25 with *p* < 0.01 between patients in acute stage of ACD than in remission; *, means statistical significance higher the percentage of CD25hCD127low with *p* values < 0.05 between patients in remission of ACD than in acute stage.

**Table 1 jcm-12-01397-t001:** Characteristic of the groups: Patients with ACD and Controls.

Parameter	Controls (*n* = 21)	ACD Patients (Acute Stage) (*n* = 26)	ACD Patients (Remission) (*n* = 26)
Age	45.76 ± 15.15	47.5 ± 14.89 NS	-
Sex [M/F]	9/12	11/15 NS	-
Exacerbation (weeks)	-	3.7 ± 3.2 (1–12)	-
Exacerbation longer than 1 week (Yes/No)	-	19/7	-
EASI Score	-	11.6 ± 7.3 (2–29.6)	-
EASI subgroup (<15/>15)	-	17/9	-
Atopic Diseases (Yes/No)	-	4/22	-
Comorbidities (Yes/No)	-	8/18	-
Allergies (Metal/Nonmetal)	-	16/10	-
WBC	6.95 ± 1.54	7.21 ± 1.38 NS	7.35 ± 1.6 NS
Lymphocytes	1.94 ± 0.56	1.69 ± 0.46 NS	1.68 ± 0.47 NS (*p* = 0.1049)
Lymphocytes (%)	25.86 ± 6.9	24.18 ± 7.02 NS	24.2 ± 7.1 NS

ACD, allergic contact dermatitis; *n*, number of patients; NS, non-significant; M/F, male/female; EASI, Eczema Area and Severity Index; WBC, white blood cells.

**Table 2 jcm-12-01397-t002:** Correlation between iOPN CD4 cells percentage with selected variables in patients with ACD.

iOPN CD4 vs:	Controls (*n* = 21)	ACD Patients (Acute Stage) (*n* = 26)	ACD Patients (Remission) (*n* = 26)
Age	−0.233 NS	0.121 NS	0.008 NS
Exacerbation (weeks)	-	−0.157 NS	−0.115 NS
EASI Score	-	0.004 NS	−0.016 NS
CD4	0.33 NS	0.174 NS	0.054 NS
CD25	0.072 NS	0.054 NS	0.229 NS
CD25h	−0.029 NS	0.213 NS	0.338 NS (*p* = 0.092)
CD25hCD127Iow	0.026 NS	0.266 NS	0.124 NS

iOPN CD4, CD4 T lymphocytes with expression of intracellular osteopontin; ACD, allergic contact dermatitis; *n*, number of patients; NS, non-significant; EASI, Eczema Area and Severity Index; CD4, CD4 positive T lymphocytes; CD25, subset of CD4 positive T lymphocytes; CD25h, T lymphocytes with high CD25 expression; CD25hCD127low, T lymphocytes with high CD25 expression and CD127 low expression—regulatory T cells. Data are presented as R values and followed with *p* values when *p* < 0.1.

**Table 3 jcm-12-01397-t003:** (**A**,**B**). Correlation between percentage of iOPN CD subpopulations in ACD patients and selected variables depending on EASI.

(**A**)		
**EASI I–iOPN CD4 vs:**	**ACD Patients (Acute Stage) ** **(*n* = 17)**	**ACD Patients (Remission)** **(*n* = 17)**
Age	0.095 NS	−0.045 NS
Exacerbation (weeks)	−0.061 NS	−0.373 NS
EASI Score	−0.057 NS	0.177 NS
CD4	0.167 NS	−0.107 NS
CD25	0.000 NS	0.32 NS
CD25h	0.301 NS	0.153 NS
CD25hCD127Iow	0.068 NS	0.352 NS
EASI I–iOPN CD4, CD4 T cells with expression of intracellular osteopontin in patients with mild to moderate eczema area and severity index (<15); ACD, allergic contact dermatitis; *n*, number of patients; NS, non-significant; EASI, Eczema Area and Severity Index; CD4, CD4 positive T lymphocytes; CD25, subset of CD4 positive T lymphocytes; CD25h, CD25 high T lymphocytes; CD25hCD127low, CD25highCD127low T cells—regulatory T cells. Data are presented as R values and followed with *p* values when *p* < 0.1.		
(**B**)		
**EASI II–iOPN CD4 vs:**	**ACD Patients (Acute Stage) ** **(*n* = 9)**	**ACD Patients (Remission)** **(*n* = 9)**
Age	0.282 NS	0.191 NS
Exacerbation (weeks)	−0.155 NS	0.205 NS
EASI Score	−0.102 NS	0.260 NS
CD4	0.102 NS	0.508 (*p* = 0.136)
CD25	−0.089 NS	−0.005 NS
CD25h	−0.023 NS	0.622 (0.061)
CD25hCD127Iow	0.481 (*p* = 0.143)	0.322 NS
EASI II–iOPN CD4, CD4 T cells with expression of intracellular osteopontin in patients with severe eczema area and severity index (>15); ACD, allergic contact dermatitis; *n*, number of patients; NS, non-significant; EASI, Eczema Area and Severity Index; CD4, CD4 positive T lymphocytes; CD25, subset of CD4 positive T lymphocytes; CD25h, CD25 high T lymphocytes; CD25hCD127low, CD25highCD127low T cells—regulatory T cells. Data are presented as R values and followed with *p* values when *p* < 0.2.		

**Table 4 jcm-12-01397-t004:** (**A**) Comparison of selected variables in ACD patients depending on presence of comorbidities. (**B**) Correlations of the percentage of iOPN CD4 lymphocytes with selected variables in the ACD patients depending on presence of comorbidities.

(**A**)					
		**Non-Comorbidities (*n* = 18)** **Acute Stage**	**Comorbidities (*n* = 8)** **Acute Stage**	**Non-Comorbidities (*n* = 18)** **Remission**	**Comorbidities (*n* = 8)** **Remission**
Age		45.06 ± 13.77	53 ± 16.78 NS		
Exacerbation (weeks)		4 (1–12)	2 (1–12) NS		
EASI Score		8.8 (2–21.9)	12.8 (5.8–29.6) NS		
CD4		42.06 ± 10.77	47.63 ± 11.69	41.78 ± 11.71	40.38 ± 11.76
CD25		5.42 ± 2.26	7.38 ± 4.95 (*p* = 0.18)	2.9 ± 1.68 ***	4.44 ± 3.31 *
CD25h		1.101 ± 0.57	1.638 ± 1.16 (*p* = 0.12)	0.86 ± 0.45	0.71 ± 0.44
CD25hCD127low		37.31 ± 15.04	35.63 ± 11.56	45.44 ± 13.59 (*p* = 0.073)	38.11 ± 16.82
*n*, number of patients; EASI, Eczema Area and Severity Index; CD4, CD4 positive T lymphocytes; CD25, subset of CD4 positive T lymphocytes; CD25h, CD25 high T lymphocytes; CD25hCD127low, CD25highCD127low T cells—regulatory T cells; NS, non-significant; *, means statistical significance higher the percentage of CD25 with *p* values < 0.05 between patients with comorbidities in acute stage of ACD than in remission; ***, means statistical significantly higher the percentage of CD25 with *p* < 0.0001 between patients without comorbidities in acute stage of ACD than in remission.					
(**B**)					
**iOPN CD4 vs:**	**Controls (*n* = 21)**	**Non-Comorbidities (*n* = 18) ** **Acute Stage**	**Comorbidities (*n* = 8) ** **Acute Stage**	**Non-Comorbidities (*n* = 18) ** **Remission**	**Comorbidities (*n* = 8) ** **Remission**
Age	−0.233 NS	0.345 NS	−0.452 NS	0.257 NS	−0.311 NS
Exacerbation (weeks)	-	−0.131 NS	−0.393 NS	−0.193 NS	−0.519 NS
EASI Score	-	−0.043 NS	0.091 NS	0.164 NS	0.033 NS
CD4	0.33 NS	0.266 NS	0.012 NS	−0.147 NS	0.807 (0.019) *
CD25	0.072 NS	−0.132 NS	0.357 NS	0.472 (0.042) *	0.355 NS
CD25h	−0.029 NS	0.11 NS	0.345 NS	0.523 (0.0342) *	0.229 NS
CD25hCD127Iow	0.026 NS	0.27 NS	0.429 NS	0.382 NS	0.012 NS
iOPN CD4, CD4 T lymphocytes with expression of intracellular osteopontin; *n*, number of patients; EASI, Eczema Area and Severity Index; CD4, CD4 positive T lymphocytes; CD25, subset of CD4 positive T lymphocytes; CD25h, CD25 high T lymphocytes; CD25hCD127low, CD25highCD127low T cells—regulatory T cells; NS, non-significant; *, means statistical significance with *p* < 0.05 between iOPN CD4 percentage and CD4 in patients with comorbidities during remission and between iOPN CD4 percentage and CD25, CD25h in patients without comorbidities during remission. Data are presented as R values and followed with *p* values when *p* < 0.05.					

**Table 5 jcm-12-01397-t005:** (**A**) Comparison of selected variables in ACD patients depending on allergen. (**B**) Correlations of the percentage of iOPN CD4 lymphocytes with selected variables in the ACD patients depending on allergen. Data are presented as R values and followed with *p* values when *p* < 0.2.

(**A**)					
		**Metals (*n* = 16)** **Acute Stage**	**Non-Metals (*n* = 10) Acute Stage**	**Metals (*n* = 16) ** **Remission**	**Non-Metals (*n* = 10) Remission**
Age		48.5 ± 14.92	45.9 ± 15.5 NS		
Exacerbation (weeks)		3.5 (1–12)	2 (1–8) NS		
EASI Score		13.25 (2–29.6)	8.8 (3.2–21.6)		
CD4		41.69 ± 9.16	47.1 ± 13.59	41.81 ± 10.88	40.6 ± 13.02
CD25		6.13 ± 3.39	5.85 ± 3.61	3.19 ± 2.35 **	3.66 ± 2.45 (0.098)
CD25h		1.33 ± 0.88	1.17 ± 0.76	0.92 ± 0.89	0.65 ± 0.37 *
CD25hCD127Iow		40.16 ± 13.38	31.4 ± 13.49	45.43 ± 16.98	39.6 ± 9.91
*n*, number of patients; EASI, Eczema Area and Severity Index; CD4, CD4 positive T lymphocytes; CD25, subset of CD4 positive T lymphocytes; CD25h, CD25 high T lymphocytes; CD25hCD127low, CD25highCD127low T cells—regulatory T cells; NS, non-significant; *, means statistical significance with *p* value < 0.05 between proportion of CD25h in patients allergic to other allergens during the acute stage than the remission; **, means statistical significance with *p* value < 0.01 between proportion of CD25 in patients allergic to metals during the acute stage than in remission.					
(**B**)					
**iOPN CD4 vs:**	**Controls ** **(*n* = 21)**	**Metals ** **(*n* = 16)** **Acute Stage**	**Non-Metals** **(*n* = 10) ** **Acute Stage**	**Metals** **(*n* = 16)** **Remission**	**Non-Metals (*n* = 10)** **Remission**
Age	−0.233 NS	−0.057 NS	0.394 NS	0.261 NS	−0.276 NS
Exacerbation (weeks)	-	−0.465 (*p*=0.052)	0.438 NS	−0.346 NS	0.436 NS
EASI Score	-	−0.014 NS	−0.296 NS	0.165 NS	−0.507 (*p* = 0.11)
CD4	0.33 NS	0.285 NS	−0.072 NS	0.251 NS	−0.352 NS
CD25	0.072 NS	0.192 NS	0.167 NS	0.304 NS	−0.025 NS
CD25h	−0.029 NS	0.146 NS	0.542 (*p* = 0.81)	0.633 (0.0062) **	−0.316 NS
CD25hCD127Iow	0.026 NS	0.251 NS	−0.046 NS	0.26 NS	0.462 (*p* = 0.13)
iOPN CD4, CD4 T lymphocytes with expression of intracellular osteopontin; *n*, number of patients; EASI, Eczema Area and Severity Index; CD4, CD4 positive T lymphocytes; CD25, subset of CD4 positive T lymphocytes; CD25h, CD25 high T lymphocytes; CD25hCD127low, CD25highCD127low T cells—regulatory T cells; NS, non-significant; **, means statistical significance correlation with *p* values < 0.01 between iOPN CD4 percentage and CD25h in patients allergic to metals examined during remission.					

## Data Availability

Data available on the request.

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
