# Peer review of "Osteopontin and Regulatory T Cells in Effector Phase of Allergic Contact Dermatitis"

_jcm, 2023, doi:10.3390/jcm12041397_

Round 1

Reviewer 1 Report

Authors wanted to determine CD4 T lymphocytes producing intracellular osteopontin (iOPN T cells) and assess the selected T lymphocyte subsets including regulatory T cells in the blood of patients with acute contact dermatitis. This article is very interesting, but requires some minor revisions. The major problem is the size of the study group which is too small for such study. There is no information how many men and women were included for this study. Also, the graphs used for the data visualisation are not clear (Fig 4 and Fig 5). 

Author Response

 Dear Reviewer,

Thank you for the time spend on the revision of our paper and for the constructive remarks. We have made the corrections that you have suggested and hope that with your help we managed to improve our manuscript. We would like to refer to each of the comments, explain and justify them.

Response to Reviewer 1 comments

Point 1. There is no information how many men  and women were included for this study.

Response 1. This information regarding number of women and men included in this study  has been corrected in main text in section results and is inserted in table 1.

Point 2. The graphs used for the data visualization are not clear.

Response 2. The graphs have been corrected.

Yours sincerely,

Authors.

Reviewer 2 Report

This manuscript discussed the mechanisms responsible for the function of osteopontin (OPN) and regulatory T cells in allergic contact dermatitis. The authors showed increased iOPN T cells in the acute ACD with a decreased percentage of regulatory T lymphocytes in the acute stage of ACD that may be related to the transformation of Tregs into CD4CD25 T cells and their increased recruitment to the skin. Also, the positive correlation between the percentage of CD4CD25 lymphocytes and the EASI index may be indirect evidence for the importance of activated lymphocytes CD4CD25 and CD8 lymphocytes as effector cells in ACD.

I have only some suggestions:

1.    The introduction section should be more concise, and many of its data should be moved to the discussion section without duplication.

2.    An ethical approval number should be added to the materials and methods section.

3.    In the result section, table 3A/3b, CD25 in acute stage p value 0.000, but authors stated NS?

4.    The reference style should be uniform.

Author Response

Dear Reviewer,

Thank you for the time spend on the revision of our paper and for the constructive remarks. We havemade the corrections that you have suggested and hope that with your help we managed to improve our manuscript. We would like to refer to each of the comments, explain and justify them.

Response to Reviewer 2 comments

Point 1. The introduction section should be more concise, and many of its data should be moved to the discussion section without duplication.

Response 1. The introduction section has been corrected.

Point 2. An ethical approval number should be added to the materials and methods section.

Response 2. Ethical approval number has been added to the materials and methods.

Point 3. In the result section, table 3A/3b, CD25 in acute stage  p value 0.000, but authors stated NS.

Response 3. The result section table 3A/3b it has been corrected.

Point 4. The reference style should be uniform.

Response 4. The references have been corrected.

Your sincerely,

Authors.

Round 2

Reviewer 1 Report

Thank you for your response. I recommend to publish the paper in present form